# Enhanced Arbovirus Surveillance with High-Throughput Metatranscriptomic Processing of Field-Collected Mosquitoes

**DOI:** 10.3390/v14122759

**Published:** 2022-12-11

**Authors:** Jana Batovska, Peter T. Mee, Tim I. Sawbridge, Brendan C. Rodoni, Stacey E. Lynch

**Affiliations:** 1Agriculture Victoria Research, AgriBio Centre for AgriBioscience, 5 Ring Road, Bundoora, VIC 3083, Australia; 2School of Applied Systems Biology, La Trobe University, Bundoora, VIC 3086, Australia

**Keywords:** mosquitoes, Culicidae, arboviruses, surveillance, metatranscriptomics

## Abstract

Surveillance programs are essential for the prevention and control of mosquito-borne arboviruses that cause serious human and animal diseases. Viral metatranscriptomic sequencing can enhance surveillance by enabling untargeted, high-throughput arbovirus detection. We used metatranscriptomic sequencing to screen field-collected mosquitoes for arboviruses to better understand how metatranscriptomics can be utilised in routine surveillance. Following a significant flood event in 2016, more than 56,000 mosquitoes were collected over seven weeks from field traps set up in Victoria, Australia. The traps were split into samples of 1000 mosquitoes or less and sequenced on the Illumina HiSeq. Five arboviruses relevant to public health (Ross River virus, Sindbis virus, Trubanaman virus, Umatilla virus, and Wongorr virus) were detected a total of 33 times in the metatranscriptomic data, with 94% confirmed using reverse transcription quantitative PCR (RT-qPCR). Analysis of metatranscriptomic cytochrome oxidase I (COI) sequences enabled the detection of 12 mosquito and two biting midge species. Screening of the same traps by an established public health arbovirus surveillance program corroborated the metatranscriptomic arbovirus and mosquito species detections. Assembly of genome sequences from the metatranscriptomic data also led to the detection of 51 insect-specific viruses, both known and previously undescribed, and allowed phylogenetic comparison to past strains. We have demonstrated how metatranscriptomics can enhance surveillance by enabling untargeted arbovirus detection, providing genomic epidemiological data, and simultaneously identifying vector species from large, unsorted mosquito traps.

## 1. Introduction

Arthropod-borne viruses (arboviruses) are distributed worldwide and, in recent years, have caused epidemics such as dengue, chikungunya, and Zika fever [1]. Dengue alone infects 390 million people a year, with a total economic cost of nearly US$40 billion [2]. Almost 30% of emerging infectious diseases are arboviral, fuelled by increasing population growth, urbanisation, globalisation and international motility [3,4]. Arboviral infections can often be asymptomatic or present with non-specific symptoms, meaning that outbreaks can go undetected until containment is no longer feasible. For instance, clinical similarity to dengue and chikungunya viral infection enabled Zika virus (ZIKV) to circulate for over a year and a half before the first detection in Brazil occurred in 2015, by which point it had already spread to over 40 countries [5]. Antibody cross-reactivity between flaviviruses and a lack of routine testing further hindered early detection [6]. Preparedness for these epidemics requires the ability to detect unexpected novel viral species and strains, and genomic information to reconstruct transmission dynamics and inform public health initiatives.

As the primary vector of arboviruses, mosquito populations are monitored by surveillance programs in order to detect and control arboviral activity. A common approach is to trap mosquitoes and test them for the presence of arboviruses using cell culture. This involves morphologically identifying the mosquitoes to species level and inoculating a homogenised subsample onto a range of suitable cell lines, which are then screened for arboviral presence using an appropriate immunological staining method or by observing for cytopathic effect [7,8]. In recent years, molecular approaches such as reverse transcription PCR (RT-PCR) have been used for arbovirus detection, with the capacity to test pools containing thousands of mosquitoes [9]. This is a significant upscale to cell culture, which loses sensitivity with pool sizes larger than 200 mosquitoes [10], resulting in only small subsamples of trap catches being tested during flood seasons when thousands of mosquitoes are trapped each week [7]. Due to the low arbovirus infection rates in mosquito populations, it is imperative to maximise sample sizes in order to increase detection probability [11]. Although RT-PCR offers sensitivity and the ability to upscale surveillance, it requires *a priori* knowledge of the virus sequence, which limits the detection of divergent strains and restricts the discovery of unexpected novel viruses. Virus-specific PCRs also limit the number of targets and can decrease in sensitivity over time due to genomic drift in rapidly evolving viruses [12,13].

Metatranscriptomics (total RNA sequencing) is an untargeted approach to virus detection that, unlike PCR, can generate whole genome sequences for all RNA viruses present in a sample. Focusing on RNA viruses is suited to arbovirus surveillance as all known arboviruses have an RNA genome, with the exception of African swine fever virus (ASFV), a double-stranded DNA virus that is transmitted by soft ticks [14]. The phylogenetic resolution offered by whole genome sequencing is particularly valuable in outbreak situations, where it can be used to reconstruct local virus transmission, elucidate the geographic origin of cases, track virus mutations, and identify highly transmissible strains [15]. For instance, genomics was used to uncover an unreported outbreak of ZIKV in Cuba and trace it to multiple introductions from other Caribbean islands, helping to direct vector control and further surveillance activities [5]. The untargeted nature of metatranscriptomics makes it ideal for arbovirus surveillance as it enables not only the detection of established viruses that can cause human disease but also novel, unexpected viruses in mosquitoes, and other organisms of interest such as parasites [16], bacteria, and fungi [17]. Furthermore, the mosquito species composition of the trap can be determined from the metatranscriptomic sequencing reads without manual sorting of the specimens [18], removing a major bottleneck in mosquito processing.

One of the challenges in implementing metatranscriptomics as a surveillance tool is the bioinformatics analysis involved in handling the sequencing data. A typical approach to metatranscriptomic virus detection involves comparing the sequencing reads or assembled contiguous sequences (contigs) to a reference database containing previously described virus sequences. Using a large, public database such as those offered by the National Center for Biotechnology Information (NCBI) provides comprehensive virus identification. However, these databases are littered with misannotated sequences, which can confound results [19]. Alternatively, smaller, curated databases can be used to achieve more trustworthy results but usually limit the scope of identification, leaving a combined approach as the best but most time-consuming option [20]. In any circumstance, the method chosen needs to be tested and standardised in order to provide reliable and consistent results.

Decreases in sequencing costs have led to a rise in metatranscriptomic studies, with mosquito pools often sequenced to characterise viromes and discover new viruses [21,22], investigate mosquito-specific virus ecology [23], and identify vector control candidates [16,24]. However, metatranscriptomics has yet to be applied to an established public health arbovirus surveillance program. As such, there is limited information on the sensitivity and specificity of metatranscriptomic arbovirus detection from mosquitoes and how this compares to established methods of arbovirus detection. A recent study introduced positive detection criteria for metatranscriptomic detection of arboviruses from pooled mosquitoes [25], but this has yet to be tested with traps containing diverse populations of mosquito and virus species.

The goal of this study is to assess the utility of metatranscriptomics in arbovirus surveillance. Using traps collected in 2016 following a significant flooding event in regional Victoria, Australia, we screened over 56,000 mosquitoes for arboviruses using metatranscriptomic sequencing. The results were confirmed using quantitative RT-PCR (RT-qPCR) and used to assess previously established positive detection criteria. The metatranscriptomic data was further utilised for mosquito and biting midge identification and detection of insect-specific viruses, both known and previously undescribed. We also compared the metatranscriptomic results to existing surveillance program data for the same traps to examine the different outcomes of established methods and novel approaches to arbovirus surveillance.

## 2. Materials and Methods

### 2.1. Mosquito Collection and Initial Arbovirus Screening

The mosquitoes used in this study were collected as part of the Department of Health Victorian Arbovirus Disease Control Program (VADCP). Mosquito collection was performed on a weekly basis using carbon dioxide-baited encephalitis virus surveillance (EVS) light traps [26] overnight at three locations in Victoria, Australia (Figure 1). The three trapping locations were in the Rural City of Mildura (Mildura) (−34.249617, 142.218261), the Shire of Gannawarra (Gannawarra) (two traps spaced 2.2 km apart; −35.707128, 143.906764 and −35.720019, 143.925958), and the Wellington Shire Council (Wellington) (−38.206653, 147.396661). One of the Gannawarra traps was positioned in bushland containing native animal hosts, whereas the other three traps were near areas populated by humans in order to sample both enzootic and epizootic zones. The traps were collected over a seven-week period during peak mosquito season from 7 November 2016 (Week 45) to 19 December 2016 (Week 51), following a significant flood event in the Murray-Darling Basin in September 2016 [27]. Upon collection, the mosquitoes from the traps were immobilised at −20 °C for 30 min and then transported to the laboratory via chilled overnight delivery.

A subsample of mosquitoes was taken from each trap for arbovirus screening by the VADCP (Appendix A). These mosquitoes were morphologically identified and screened for arboviruses using a cell culture-based system as previously described [28]. The remaining mosquitoes from each trap were stored at −20 °C until used in this study for metatranscriptomic sequencing.

### 2.2. Sample Preparation and Nucleic Acid Extraction

Each mosquito trap was sorted into different-sized subsamples. The first subsample for each trap consisted of 100 mosquitoes, which were counted and weighed. The remainder of each trap was then sorted into subsamples consisting of 1000 mosquitoes based on the weight of the initial 100 mosquitoes. This sample size was chosen based on previous findings indicating metatranscriptomic sequencing can detect an arbovirus from a single infected mosquito in a pool of 1000 [25]. Any remaining mosquitoes were weighed and allocated as the last “remainder” subsample for each trap (estimated range: 78 to 906 mosquitoes). All mosquito subsamples were placed into 50 mL Falcon tubes and stored at −80 °C until further use. A total of 86 subsamples from 21 traps were prepared.

To homogenise the mosquitoes for nucleic acid extraction, 3 mm glass beads (Sigma-Aldrich, St. Louis, MO, USA) and Buffer AVL (Qiagen, Hilden, Germany) scaled according to mosquito number (Appendix A) were added to each subsample. The mosquitoes were then mechanically homogenised for two 1 min cycles at 1200 rpm using a 2010 Geno/Grinder (SPEX SamplePrep, Metuchen, NJ, USA). The homogenised mosquitoes were centrifuged for 5 min at 15,344× *g*, and 140 μL of supernatant was removed from each subsample. Nucleic acid was extracted from the supernatant using the QIAamp Viral RNA Mini Kit (Qiagen, Hilden, Germany) according to the manufacturer’s instructions, excluding the addition of carrier RNA. A double elution was performed using 2 × 40 μL of Buffer AVE and stored at −80 °C until library preparation. An extraction negative control was included in each batch of extractions and consisted of 140 μL of Buffer AVL as input.

### 2.3. Metatranscriptomic Sequencing

Libraries were prepared for metatranscriptomic sequencing using the NuGEN Ovation Universal RNA-Seq System with custom rRNA depletion. Briefly, 2 μL of undiluted RNA (68–200 ng) was DNase treated, converted into cDNA, and sheared into 200–400 bp fragments using a Covaris S220 Focused-ultrasonicator. A library negative control was included in each batch of library preparation and consisted of 2 μL of UltraPure water (Invitrogen, Waltham, MA, USA) as input. Additionally, 2 μL of RNA from a pool of 1000 mosquitoes containing a single Ross River virus (RRV) infected mosquito [25] was used for a positive control library. After performing end repair, adaptor ligation and strand selection, the libraries were depleted of mosquito rRNA sequences using customised probes [25]. Libraries underwent 14 cycles of PCR amplification and were then purified using AMPure XP beads (Beckman Coulter, Brea, CA, USA). The libraries were quantified using a D1000 ScreenTape with the 2200 TapeStation (Agilent Technologies, Santa Clara, CA, USA) and a dsDNA HS assay with the Qubit 3 Fluorometer (Thermo Fisher Scientific, Waltham, MA, USA) and pooled into three samples of equimolar concentration. Each pooled sample was quantified using the TapeStation, diluted to 20 nM with UltraPure water, and treated with Free Adapter Blocking Reagent (Illumina, San Diego, CA, USA) according to the manufacturer’s instructions in order to reduce index hopping [29]. The treated sample pools were quantified using the Qubit, diluted to 10 nM with 10 mM Tris buffer (pH 7.0; Invitrogen, Waltham, MA, USA) and sequenced on a lane of the HiSeq 3000 (Illumina, San Diego, CA, USA) with 2 × 150 bp reads.

### 2.4. Read Assembly and Taxonomic Classification

The reads were demultiplexed into subsamples and used for *de novo* assembly, performed using Trinity v2.4.0 [30] with read normalisation and trimming options selected. Assembled contigs over 500 bp [18] were taxonomically classified using DIAMOND BLASTx v0.9.22.123 [31] with the NCBI non-redundant (nr) protein database (acquired 2 September 2019) and an e-value threshold of 10^−5^. Abundance was measured by mapping trimmed, interleaved reads back to the contigs using BWA-MEM v0.7.17-r1188 [32] and obtaining read counts with SAMtools v1.9 idxstats [33]. Taxonomy, abundance and sample information were imported into RStudio v1.2.1335 [34] for analysis and visualisation with phyloseq v1.28.0 [35] and ggplot2 v3.2.1 [36] packages. The abundance data were normalised to even sampling depth, and taxa with a mean abundance lower than 10^−5^ were removed [37]. Non-metric multidimensional scaling (NMDS) based on Bray–Curtis dissimilarity was used to compare taxa in the trap subsamples, the positive control, and the extraction and library negative controls.

### 2.5. Metatranscriptomic Mosquito and Biting Midge Species Identification

The mosquito (Diptera: Culicidae) and biting midge (Diptera: Ceratopogonidae) species in each trap were determined by using BLASTn v2.9.0+ [38] to compare the subsample contigs to a custom database of cytochrome oxidase I (COI) barcode sequences. The database contained 138 COI sequences belonging to 29 mosquito species [39] and 13 biting midge species [40] found in Victoria, Australia. Biting midges were included in the database as they are often found in mosquito traps and can also transmit arboviruses [41]. Members of the *Culex pipiens* mosquito species complex that cannot be differentiated by COI had their names conglomerated so that they would be counted as one [39]. The database is accessible via Figshare: https://doi.org/10.6084/m9.figshare.10246826.v3 (accessed on 31 March 2020). The BLASTn was performed with an e-value threshold of 10^−5^, and the results were filtered for contigs with >300 bp alignment length and >95% identical match to COI sequences. Read counts for these contigs were acquired from the previously performed idxstats analysis, summed per species and per trap, and used to plot species abundance with ggplot2.

### 2.6. Targeted Arbovirus Screen

The metatranscriptomic data generated from the trap subsamples were screened for arboviruses of public health interest using a targeted custom database based on those listed in Mackenzie et al. [42] and Vasilakis et al. [43]. Whole genome sequences were used in the database if publicly available, and segmented genomes were merged so that each arbovirus was represented by a single sequence. The resulting database contains 74 arboviruses representing nine viral families and is available on Figshare: https://doi.org/10.6084/m9.figshare.12055830 (accessed on 31 March 2020). The screening was performed by mapping the subsample contigs and reads to the arbovirus database with BWA-MEM and using BBMap pileup [44] to measure the average Fold Coverage by Reads (FCR), Percent Coverage by Contigs (%CC), and Percent Coverage by Reads (%CR). Each arbovirus coverage value in the subsamples was divided by any corresponding coverage in the negative controls, resulting in three coverage-based criteria: Fold Coverage by Reads ratio (FCR-r), Percent Coverage by Contigs ratio (%CC-r), and Percent Coverage by Reads ratio (%CR-r), with values ≥ 2 considered positive [25]. The performance of these criteria was compared by confirming every detection with RT-qPCR (as outlined below). Lastly, the SAMtools idxstats command was used to get read counts for the arboviruses in each trap based on the read alignments.

### 2.7. Confirmation of Arbovirus Detections Using RT-qPCR

RT-qPCR was used to test all 86 subsamples for any arboviruses detected during the targeted arbovirus screen, with a total of five assays performed: Ross River virus (RRV), Sindbis virus (SINV), Trubanaman virus (TRUV), Umatilla virus (UMAV) and Wongorr virus (WGRV). The mosquito subsample RNA was diluted 1:5 with UltraPure water (Invitrogen, Waltham, MA, USA) for use in the RT-qPCR assays. The RRV [45], SINV [46], TRUV [28] and UMAV [47] primers were previously published, and the WGRV primers were designed in-house. Further details on the primers, as well as the PCR cycles and kits used, can be found in Appendix A. For the probe-based RRV assay, subsamples with Ct < 40 were considered positive; for the SYBR Green-based SINV, TRUV, UMAV and WGRV assays, subsample melt curves were also compared to negative and positive control melt curves to determine positivity.

### 2.8. Phylogenetic Analysis of Target Arboviruses

In order to perform phylogenetic analysis, consensus sequences were derived from the read-based alignments that had >90% coverage of reference sequences from the targeted arbovirus screen. SAMtools mpileup and BCFtools consensus were used to generate the consensus sequences, and then MAFFT v7.429 [48] was used to align them with other arbovirus sequences from GenBank. Full genome alignments were used for RRV and SINV (11,362 nt and 11,460 nt, respectively), Segment M for TRUV (4152 nt), and Segment 7 for UMAV (1364 nt). Phylogenetic trees were created for each arbovirus alignment with PhyML v3.3 [49] using maximum likelihood and a general time reversible (GTR) substitution model. Branch support was evaluated using 1000 bootstrap replicates. The resulting trees were viewed and edited using Geneious v8.1.8 [50]. All arbovirus sequences generated for phylogenetic analysis have been uploaded to GenBank (acc. OP950205–OP950214).

### 2.9. Virome Analysis

Contigs from the initial *de novo* assembly classified as viral by DIAMOND BLASTx were used to investigate the broader virome present in the trap subsamples. The contigs were compared to the NCBI nucleotide (nt) database (acquired 28 October 2019) using BLASTn with an e-value threshold of 10^−5^ and filtered to remove contigs with non-viral matches. The remaining viral contigs were further filtered for matches to RNA viruses by comparing them to all viral RNA-dependent RNA polymerase (RdRp) protein sequences on RefSeq using DIAMOND BLASTx with an e-value threshold of 10^−5^.

Abundance estimates were determined by mapping reads back to the RNA virus contigs and measuring read counts, as previously described, and summing the reads per trap. The RNA virus contig abundances were imported into phyloseq along with COI contig abundances (as previously identified via the COI BLASTn search). The read counts were normalised and filtered to remove taxa with a mean abundance below 10^−5^ [37]. The difference in viral and mosquito taxa amongst the subsamples was then visualised using an NMDS based on Bray–Curtis dissimilarity.

The RNA virus contig RdRp BLASTx results and associated abundance information were plotted in RStudio using ggplot. For the sake of brevity, only RNA virus contigs greater than 1000 bp in length [51], with at least 100 reads and over 500 bp alignment length, were included in the figure. Investigation of possible redundancies in virus detection due to highly similar RdRp references was carried out by comparing all matching GenBank RdRp protein sequences from the BLASTx results using a maximum likelihood tree, as previously described.

## 3. Results

### 3.1. Sample Preparation and Sequencing

Based on weight, a total of 62,218 mosquitoes were trapped in Gannawarra, Mildura and Wellington over a seven-week period in 2016 (Weeks 45–51; Figure 2 and Appendix A). The VADCP subsampled 5985 mosquitoes (9.6%) to use for cell culture-based arbovirus screening, with the remaining 56,233 mosquitoes (90.4%) used for metatranscriptomic sequencing. Mosquitoes were sorted into 2–15 subsamples per trap, resulting in a total of 86 mosquito subsamples for sequencing (Appendix A). Additionally, there were three extraction negative controls, four library negative controls, and one positive control, resulting in a total of 94 samples for sequencing.

For the 86 mosquito subsamples, a mean of 10,575,201 paired reads was generated per subsample (range: 7,971,017–16,414,900). When the reads were mapped to taxonomically classified contigs (Figure 3), a mean of 78% (range: 17–99%) belonged to eukaryotes, the majority of which were arthropod species (mean: 70%; range: 12–98%). A substantial proportion of the reads (mean: 61%; range: 4–95%) were attributed to two ciliate species (*Oxytricha trifallax* and *Stylonychia lemnae*) and two nematode species (*Wuchereria bancrofti* and *Brugia timori*). However, further investigation indicated that these reads were derived from mosquito rRNA, and so they were re-classified as arthropod for counting. Archaea and bacteria were represented by a mean of 1% of reads (range: 0.0–17%) and were not characterised as part of this study. The percentage of viral reads varied among the subsamples (1–82%), with certain traps having more viral reads than others. For instance, in Gannawarra, the Week 45 and 46 traps had a mean of 55% viral reads, whereas the Week 47–51 traps had a mean of 8% viral reads.

The three extraction negative controls had a low number of sequencing reads compared to the subsamples, with a mean of 8112 paired reads per sample (range: 5249–10,139). Three out of the four library negative controls also had a low number of sequencing reads (mean: 112,252; range: 5245–258,192). However, there was one library negative control with 11,926,882 paired reads. Of the taxonomically classified reads for this library negative control, 80% were plant and 16% were bacterial, suggesting contamination. These plant and bacterial species were also present in some other samples from the same sequencing pool as the contaminated library negative control but not in samples that were processed with it during library preparation, indicating that contamination occurred during sequencing. When comparing all taxa, both the extraction and library negative controls were distinct from the trap subsamples (Appendix A). The taxonomic composition of the positive control was similar to the subsamples.

### 3.2. Mosquito and Biting Midge Species Identification

The COI-based analysis of the metatranscriptomic data from all 86 subsamples identified 12 mosquito and two biting midge species in the 21 traps used in this study. The 12 mosquito species were detected over the seven-week period, with the two inland locations, Gannawarra and Mildura, sharing similar mosquito species in contrast to the coastal location, Wellington, which primarily had *Aedes camptorhynchus* (Figure 4A). The mosquito species composition changed over time, particularly in Gannawarra, where *Anopheles annulipes* and *Culex australicus*/*globocoxitus* populations were gradually replaced by *Culex annulirostris*. The two biting midge species were *Culicoides marksi* in Mildura and *Culicoides multimaculatus* in Wellington, both detected at low abundances (<5% of mosquito and midge COI trap reads).

When only the 100-mosquito subsamples were used for COI-based metatranscriptomic analysis, abundance estimates for prevalent species were often comparable to those generated with all of the trap subsamples (Figure 4B). However, only 11 of the 14 species were present in the 100-mosquito subsamples, with low abundance (<5%) mosquito species *Aedes theobaldi* and *Tripteroides atripes*, and biting midge species *Culicoides marksi,* not detected. Analysis using only the 100-mosquito subsamples also resulted in the taxonomic dropout of other species, such as *Anopheles annulipes*, which was absent in seven of 19 traps positive for this species. In total, there were 26 taxonomic dropouts in 16 of the 21 traps when using only the 100-mosquito subsample data, compared to when all of the trap subsamples were used.

Relative abundance estimates based on morphological identification of initial trap subsamples by the VADCP in 2016 were similar to COI-based metatranscriptomic estimates for the most prevalent species, despite representing two separate mosquito subsamples (Figure 4C). However, there were differences in the detection of lower abundance species, with morphology-based methods identifying 8 of the 14 species detected using metatranscriptomics. It should be noted that the VADCP does not identify or record biting midge species present in the traps, which accounts for two of the undetected species.

### 3.3. Assessment of Positive Detection Criteria

Three criteria were assessed for arbovirus detection during the targeted screen of the metatranscriptomic data: Fold Coverage by Reads ratio (FCR-r), Percent Coverage by Contigs ratio (%CC-r), and Percent Coverage by Reads ratio (%CR-r), with values ≥ 2 considered positive [25]. Counting at the trap level, a total of 9 detections were made using FCR-r, 15 using %CC-r, and 22 using %CR-r (Appendix A). No detections were made in the negative controls. All trap detections were confirmed using RT-qPCR, making %CR-r the most effective criterion to use for the targeted arbovirus screen of the metatranscriptomic data.

### 3.4. Arbovirus Detection

Using the targeted database with the %CR-r criterion to screen the metatranscriptomic data resulted in the detection of five arboviruses: RRV, SINV, TRUV, UMAV and WGRV (Figure 5A). Out of the 86 subsamples, 25 were positive for one or more arbovirus, resulting in 33 detections. Counting at trap level, these represented 22 detections in 13 of the 21 traps. The majority of the trap detections came from Mildura (54.6%) and Gannawarra (31.8%), with only 13.6% from Wellington. RRV was detected in all three locations, SINV, TRUV and WGRV in Gannawarra and Mildura, and UMAV in Mildura. Based on read number per trap, the most highly abundant arboviruses were SINV (mean 998.7 reads), TRUV (mean 663.8) and RRV (mean 514.5), whereas fewer reads were attributed to UMAV (mean 137.7) and WGRV (mean 3). Out of the 33 detections, 22 (67%) were in a 1000-mosquito subsample, nine (27%) in a remainder subsample (509–799 mosquitoes), and two (6%) in a 100-mosquito subsample, both of which were TRUV (Figure 5B).

Of the 33 metatranscriptomic subsample detections, 31 were confirmed using RT-qPCR (Appendix A). The two unconfirmed detections were both SINV in Gannawarra from traps that had other subsamples positive for SINV via metatranscriptomics. The RT-qPCR testing revealed an additional 12 detections in the subsamples or counting at trap level, an additional four detections: one RRV, two TRUV, and one WGRV (Appendix A). Based on the RRV RT-qPCR results, lower Ct values corresponded to higher %CR (R^2^ = 0.9, Appendix A). RRV-positive subsamples with a Ct < 30 had %CR > 98, whereas subsamples with a Ct > 35 had %CR < 10. A quarter (25.6%) of all the qPCR subsample detections had a Ct > 35 (Appendix A).

Despite representing a separate subsample of mosquitoes in each trap, the initial cell culture screening performed by the VADCP detected four out of the five arboviruses detected via metatranscriptomics (Figure 5C). UMAV was the only virus not detected by the VADCP in these traps. However, the VADCP orbivirus (UMAV and WGRV) screening was not as extensive as for other viruses of public health significance due to orbivirus isolates producing cytopathic effects in mosquito cells but not in mammalian cells [28].

### 3.5. Phylogenetic Analysis of Arboviruses

Out of the 33 metatranscriptomic arbovirus detections made in the subsamples, only 10 provided the coverage required to generate a consensus sequence for use in phylogenetic analysis: three for RRV (whole genome, mean 11,888.3 nt); three for SINV (whole genome, mean 11,610.3 nt); three for TRUV (Segment M, mean 4343 nt); and one for UMAV (Segment 7, 1342 nt).

All three RRV detections grouped within the recently described Genotype 4 (G4) lineage (Figure 6A), to which all contemporary RRV strains belong [52]. The Gannawarra and Mildura RRV detections were placed in the G4A sublineage, which contains mosquito-derived strains from Queensland (QLD) and Western Australia (WA) and human-derived strains from QLD. The Wellington RRV detection clustered with the smaller G4B sublineage, which consists of mosquito-derived strains from WA and human-derived strains from QLD. All three RRV detections shared > 98% nucleotide identity with the G4 strains.

Phylogenetic analysis revealed that all three SINV detections belonged to the SINV-II genotype (Figure 6B), along with Australian strain 18953, which was isolated from *Culex annulirostris* mosquitoes in 1975, and Chinese strain YN_222, which was isolated from a midge in 2013 [53]. SINV from both Gannawarra and Mildura shared 96% nucleotide identity with Australian strain 18953 and 90% with Chinese strain YN-222. The other Australian SINV strain, SW6562, which was isolated in 1984 and belongs to the SINV-VI or the Southwest Australia genotype [54], shared 72% nucleotide identity with SINV from the Gannawarra and Mildura traps. Interestingly, SINV from one of the Gannawarra subsamples shared more nucleotide identity with SINV from the Mildura subsample (99.51%) than with SINV from the other Gannawarra subsample (99.16%), despite belonging to the same trap. 

When compared with Segment M sequences from orthobunyaviruses in the Mapputta group, the three TRUV detections clustered with other TRUV strains (Figure 6C), forming a clade with the type strain MRM3630, isolated in QLD in 1965, strain SW27572, isolated in WA in 1993, and strain Murrumbidgee 934 (also known as Murrumbidgee virus), isolated in New South Wales (NSW) in 1997 [55,56,57]. These three TRUV strains were all isolated from *Anopheles annulipes* mosquitoes. The TRUV detections from Gannawarra in Week 46 and Mildura in Week 51 were similar (99.47%) and shared > 98% nucleotide identity with strains SW27572 and Murrumbidgee 934. The TRUV from Mildura in Week 47 was 5% different to the other two detections, sharing most nucleotide identity with strain MRM3630 (97.3%).

Of the three UMAV detections from Mildura, only the one from Week 50 had enough coverage of a genome segment to allow phylogenetic analysis (Figure 6D). Comparison of Segment 7 showed the Mildura UMAV detection was most similar to UMAV M4941_15 (94.19% nucleotide identity), which was isolated from *Culex quinquefasciatus* mosquitoes in 2015, also in Victoria [25]. The two Victorian strains grouped with the Japanese Koyama Hill virus (KHV) [58], forming a separate clade to the two American UMAV strains.

### 3.6. Virome Ecology

In addition to the five arboviruses detected using the targeted database, contigs matching 51 other viruses were assembled from the trap subsample reads, ranging in size from 1–20 kbp (Figure 7). Of the 51 viruses, 32 are from an existing viral group, with a total of 16 viral families or orders represented, whereas the other 19 viruses are currently unclassified. Some of the viruses are closely related (Appendix A) and likely belong to the same viral species. All of the viruses are insect-specific or have no known vertebrate host, except for Hypsignathus monstrosus dicistrovirus (HMDV), which was sequenced from fruit bats but originated in arthropods [59], and Fisavirus 1 (FSV1), which was sequenced from the intestinal content of freshwater carp and is also of arthropod origin [60]. Furthermore, the contigs matching HMDV and FSV1 had a mean 50% and 38% RdRp protein similarity to the reference sequences, respectively, which is indicative of new species [51]. Almost half of the viruses (25 out of 51) shared < 90% RdRp protein similarity with the matching contigs and likely represent novel viral taxa. The longest contig for each virus has been made available (https://doi.org/10.6084/m9.figshare.17071970, accessed on 20 July 2022).

Based on the read number per positive trap, the most abundant viruses were those matching Hubei arthropod virus 1 (mean 2,775,315.8 reads), Ngewotan virus (1,063,130.1), Culex-associated Tombus-like virus (419,002.7), and Yongsan picorna-like virus 2 (313,737.5). Certain viruses fluctuated in abundance over time, such as Ngewotan virus [61], which in Gannawarra went from a mean of 2,567,730 reads in Weeks 45 to 48 to a mean of 34,499 reads in Weeks 49 to 51. This pattern of abundance for Ngewotan virus was repeated in Mildura but not in Wellington. Other viruses also had location-specific patterns of abundance, with 15 of the 51 viruses detected in only one of the three locations (Figure 7).

Differences in virus abundance and geography were influenced by the mosquito species present in the traps. For instance, the presence of viruses such as Aedes camptorhynchus reo-like virus in only Wellington (Figure 7) can be related to the abundance of *Aedes camptorhynchus* mosquitoes in traps from that location (Figure 4A). When the virus and mosquito species were compared in the three locations, the species found in inland Gannawarra and Mildura were similar, in contrast to the species found in coastal Wellington (Figure 8). Furthermore, the virus and mosquito species found in Gannawarra and Mildura varied over time, whereas in Wellington, there was minimal variation in the species detected over the seven-week trapping period. For each trap, the species composition was largely similar amongst the subsamples, with outliers often a result of mosquito number variations (Appendix A).

## 4. Discussion

We demonstrate the utility of metatranscriptomics as a high-throughput arbovirus surveillance tool by screening over 56,000 mosquitoes from 21 traps and detecting five arboviruses of public health interest. Additionally, the metatranscriptomic data was used to determine the species composition of the traps and to survey the broader viral diversity, highlighting the versatility of the data.

The metatranscriptomic COI analysis resulted in the detection of 12 mosquito species and two biting midge species in the traps (Figure 4A). Whilst COI could not differentiate certain species (i.e., members of the *Culex pipiens* complex), it detected others that are difficult to identify morphologically, such as *Culex palpalis*, which is almost indistinguishable from the closely related *Culex annulirostris*, an important Australian vector species [62]. Furthermore, by detecting biting midge species such as *Culicoides marksi*, a vector of animal arboviruses [41], metatranscriptomics further extends the utility of surveillance to veterinary health. Whereas morphological identification of a separate insect family requires extensive taxonomical expertise, metatranscriptomic identification only requires the addition of reference sequences to the database used during analysis, thereby enabling the detection of any species with a distinct, curated barcode sequence. Continued efforts to grow comprehensive barcode databases based on accurately identified specimens for species relevant to surveillance are essential in broadening the capacity of metatranscriptomic species identification [63] (Table 1).

Targeted screening of the metatranscriptomic data resulted in the detection of five arboviruses relevant to public health: RRV, which causes notifiable disease in humans and animals [64]; SINV and TRUV, which have been linked to human and animal arboviral infection [54,65,66]; and UMAV and WGRV, which are both serologically linked to infections in animals [67,68]. The coverage provided by metatranscriptomic sequencing enabled phylogenetic analysis using long stretches of the viral genomes (range: 1364–11,460 nt), offering valuable insights into genotypic diversity, viral lineages, and geographical differences. For instance, whole genome phylogenies revealed the presence of two geographically separated RRV sublineages in Victoria (Figure 6A), with the inland Gannawarra and Mildura RRV detections placed within the G4A sublineage and the coastal Wellington RRV detection placed within the G4B sublineage [52]. The distinct lineages warrant monitoring as ongoing evolution may lead to changes in RRV fitness and virulence [69]. As for SINV, it is unclear why the detections in two subsamples from the same Gannawarra trap were less similar than the SINV detected in Mildura (Figure 6B). One possible explanation is that the detections came from different mosquito species, with all three subsamples containing *Culex annulirostris* and *Culex australicus*/*globocoxitus* mosquitoes, both of which have been implicated in SINV transmission [54]. These results demonstrate how splitting traps into subsamples for screening helps to uncover the viral diversity present within trap mosquito populations while highlighting a need for more extensive sampling to better understand the genetic diversity of SINV in Victoria. We have made available all of the arbovirus sequences used for phylogenetic analysis, which will strengthen future analyses.

More publicly available reference material could have improved the analysis of the detected orbiviruses (UMAV and WGRV). Phylogenetic analysis of the NS2 protein (Segment 7) from UMAV showed that it was most similar to another UMAV sequence from Victoria [25] and KHV from Japan [58], together forming a clade separate to two UMAV strains from the USA (Figure 6D). Though useful to know these relationships, it is a limited representation of the *Umatilla virus* species, which also contains Stretch Lagoon orbivirus (SLOV), Minnal virus (MINV), Netivot virus (NETV), and Llano Seco virus (LLSV) [67]. These viruses do not have available reference sequences, apart from SLOV, which has sequences for all segments except for 3, 7, and 10 [47,70]. The issue of limited reference material was particularly evident with WGRV, which only had 1368 bp of genomic sequence available for inclusion in the Australian arbovirus reference database used to screen the trap subsamples (https://doi.org/10.6084/m9.figshare.12055830, accessed on 31 March 2020) and insufficient coverage (<30%) to perform phylogenetic analysis. In an attempt to acquire more of the WGRV genome, the assembled contigs were screened for sequences similar to those publicly available for WGRV, but none were found (unpublished data). The search was confounded by the segmented nature of the WGRV genome and the presence of other orbiviruses in the same traps (Figure 5A and Figure 7). Evidently, the utility of metatranscriptomic surveillance is dependent on the availability of genomic reference material, stressing the importance of sequencing archival, curated arbovirus collections [71] (Table 1). Further attempts to acquire the entire WGRV genome from mosquito homogenate could utilise cell culture, which can increase the viral titre to facilitate whole genome sequencing [56].

The majority of metatranscriptomic arbovirus detections were confirmed by RT-qPCR, with only two subsample detections missed by the SINV assay (Appendix A). The four metatranscriptomic SINV subsample detections that were confirmed by RT-qPCR all had high Ct values (>38), even though three of them had high %CR values (>98) via metatranscriptomic sequencing, which typically corresponds to low Ct values (Appendix A). It is likely primer inefficiency is responsible for the lower sensitivity and high Ct values of the SINV PCR assay. The SINV primers used were designed to detect a broad range of alphavirus species, and only the SINV-I genotype was included in the primer design [46], possibly missing important differences in the primer region present in the SINV-II genotype that was detected in the trap subsamples. On inspection, there were four and six mismatches within the 24 and 28 base SINV primers used, respectively, when compared to the SINV genomes assembled from the subsample metatranscriptomic data (unpublished data). Primers that are designed to amplify a broad range of targets are known to have lower detection sensitivity [72], especially when dealing with complex sample types such as bulk-homogenised mosquitoes, which can contain PCR inhibitors [73]. The untargeted approach of metatranscriptomics means differences in arbovirus genotypes do not affect detection, providing enhanced surveillance capabilities.

Apart from SINV detection, screening with RT-qPCR offered highly sensitive results, with an additional four arbovirus detections made at a trap level: 1 RRV, 2 TRUV and 1 WGRV. The RRV detection had a Ct value of 39.82, indicating RRV was present at a very low concentration and at the limit of detection [45]. These results are consistent with other studies that have shown RT-qPCR is more sensitive than metatranscriptomics in detecting specific viruses from complex sample types [74,75,76]. However, it is misleading to compare the two technologies on only this measure, considering that metatranscriptomics also detects an abundance of other viruses and also the host species, all within the single sequencing reaction. As for TRUV, one of the qPCR detections was also detected via metatranscriptomics but had a %CR of 1.52, which fell below the positive detection threshold of ≥2. This was true of two other subsample detections (Appendix A) and may be reason to further optimise the positive detection threshold of %CR for metatranscriptomic detection of viruses. Finally, the missed WGRV detection is likely affected by the lack of whole genome reference sequences, as discussed previously. The use of short sequences for screening lowers the chance of detection, particularly if the virus is of low abundance, highly divergent, or has a segmented genome [77]. The utility of genomic surveillance will increase over time as more whole genome sequences of local arboviruses are made available (Table 1).

The metatranscriptomic results were supported by the initial screening performed by the VADCP during the 2016–2017 surveillance season, which recorded similar mosquito species and arbovirus detections, despite using a separate subsample of mosquitoes in each trap (Figure 4C and Figure 5C). The use of labour-intensive techniques, such as morphological mosquito identification and cell-culture-based arbovirus detection, meant that during the surveillance season, the VADCP screened just under 6000 mosquitoes (9.6%) from the 21 traps used in this study, which contained over 62,000 mosquitoes in total. The high-throughput capabilities of metatranscriptomics allowed screening of the other 90.4% of mosquitoes, amounting to over 56,000 mosquitoes in total, making this the largest mosquito metatranscriptomic study to date. In addition to mosquito species identification and arbovirus detection, metatranscriptomics was able to provide additional surveillance information, including biting midge identification, genomic information for arboviruses, and a profile of the trap virome, all within a single reaction. The metatranscriptomic laboratory protocol required one technician and took approximately seven days: three days for sample preparation and four days for sequencing using the Illumina HiSeq. The processing time could be considerably reduced by automating parts of the sample preparation and utilising a different sequencer, such as the Illumina NovaSeq, which halves the sequencing time to two days [78]. The cost to metatranscriptomically screen each subsample containing up to 1000 mosquitoes was approximately AUD$230, not including labour. Switching to in-house rRNA depletion and the NovaSeq would reduce this to approximately AUD$110 (Table 1). As high-throughput sequencing continues to decrease in price and turnaround time, metatranscriptomics will progressively become the most cost-effective option for arbovirus surveillance.

The subsampling employed by the VADCP in response to surges in mosquito numbers during the 2016–2017 surveillance season (Appendix A) assumes that the mosquitoes and arboviruses present in one subsample will be indicative of the whole trap. To investigate if this was true using metatranscriptomic sequencing, an additional analysis using only 100mosquito subsamples from each trap was performed. While the COI-based species composition was predominantly the same as when all trap subsamples were used (Figure 4B), the arbovirus detections were greatly reduced, with only two metatranscriptomic arbovirus detections made using the 100-mosquito subsamples, compared to 33 detections made using all of the trap subsamples (Figure 5B). These results are reflective of the low arbovirus infection rates in mosquito populations and support sequencing all trapped mosquitoes to maximise the probability of detection [11] (Table 1). Future research could investigate the effect of homogenising the whole trap and subsampling this for surveillance.

Metatranscriptomic arbovirus detection was determined by the percent coverage of the arbovirus genome by reads ratio (%CR-r), which was shown to be the most effective criterion for detection from the field traps (Appendix A). This contrasts previous research that used the percent genome coverage by contigs ratio (%CC-r) and average fold genome coverage by reads ratio (FCR-r) as criteria for positive detection of arboviruses, based on pooled mosquito samples spiked with known concentrations of arboviruses [25]. The differences in criteria performance may partly be explained by the low abundance of arboviruses in some of the field traps (Figure 5A) compared to the high titres used in the spiking study. Differences in abundance may have also been due to the degradation of viral RNA in the field traps, compared to the viral spikes grown via cell culture [79,80]. Low arbovirus abundance hindered contig assembly and prevented depth of coverage, thereby lowering the sensitivity of %CC-r and FCR-r, respectively, rendering them less useful when applied to field traps (Appendix A). Unlike the spiking study, the low arbovirus abundance also meant there were no arbovirus reads in the negative controls as a result of index cross-talk [81], thereby negating the need for normalisation of arbovirus coverage metrics and changing the efficacy of the criteria. As such, the higher sensitivity offered by the %CR-r criterion makes it the most suitable for arbovirus detection from field traps. Further testing of the criterion using statistically robust methods is encouraged to validate its usage in routine surveillance activities [77] (Table 1).

Further analysis of the metatranscriptomic trap data revealed a broad viral diversity, with assembled contigs matching 51 viruses specific to or originating from arthropods (Figure 7). This is the first time the viral diversity of mosquitoes from Southeast Australia has been explored using a metatranscriptomic approach. Based on amino acid similarity (range: 31.3–100%), some of the detected viruses are novel and warrant further investigation. Insect-specific viruses can assist in understanding virus evolution [82], be applied to vaccine production and diagnostics [83], and have potential as biocontrol agents [84]. Future efforts in these areas will be supported by the representative viral sequences provided by this study. These sequences can also assist routine surveillance activities by enabling the identification of sequences belonging to the endemic mosquito virome and reducing viral dark matter, which improves the efficiency of detecting unexpected or emerging viruses [85].

While the exploration of broader viral diversity in the field traps enabled the detection of both known and novel viruses, it was also a complex and time-consuming process that would challenge routine surveillance activities. Unlike the targeted arbovirus screening of the metatranscriptomic data, there are no established criteria for the positive detection of previously undescribed viruses. While there are standards for reporting metatranscriptomic virus genomes [86], it is difficult to classify divergent viruses that often only have partially assembled genomes, let alone establish robust detection criteria. Determining whether a contig sequence is divergent enough from known viruses to constitute a new taxon is dependent on guidelines that vary between different viral groups [87]. This is further complicated by a lack of formal taxonomic classification for many of the viruses derived from metatranscriptomic sequencing. Often these unclassified viruses have been sequenced from invertebrate samples and are the closest match to viral sequences generated from mosquito samples (Figure 7). Unclassified viruses hinder the efficiency of metatranscriptomic arbovirus surveillance because they require further investigation to determine if their detection is of significance to public health, which typically involves lengthy phylogenetic analyses [88,89]. Metatranscriptomic arbovirus surveillance would benefit from the development of an analysis tool that would automate the process of determining the public health risk associated with novel or unclassified viral sequences detected in field traps (Table 1).

By processing unsorted, bulk mosquito traps for untargeted, high-throughput arbovirus detection and vector species identification, we have demonstrated metatranscriptomics as a high-value resource for arbovirus surveillance programs. The methods and resources presented here, including the curated reference sequence databases and refined positive detection criteria, can help facilitate the incorporation of metatranscriptomics into routine surveillance activities. Future efforts should focus on standardising operating procedures, further refining limits of detection to diagnostic standards, optimising the protocol to lower assay cost, developing user-friendly data analysis software, and expanding reference sequence databases (Table 1). The implementation of mosquito-based metatranscriptomic arbovirus detection will herald a new era of genomic surveillance that strengthens our ability to detect, track, and contain arboviral outbreaks and improve public health.

## Figures and Tables

**Figure 1 viruses-14-02759-f001:**
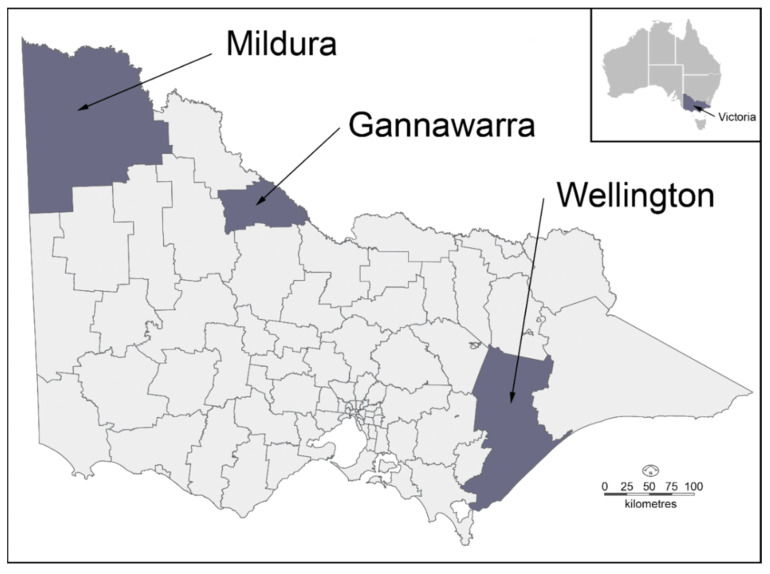
A map of the three locations traps were set up in Victoria, Australia. The highlighted regions represent the local government areas of Mildura, Gannawarra and Wellington.

**Figure 2 viruses-14-02759-f002:**
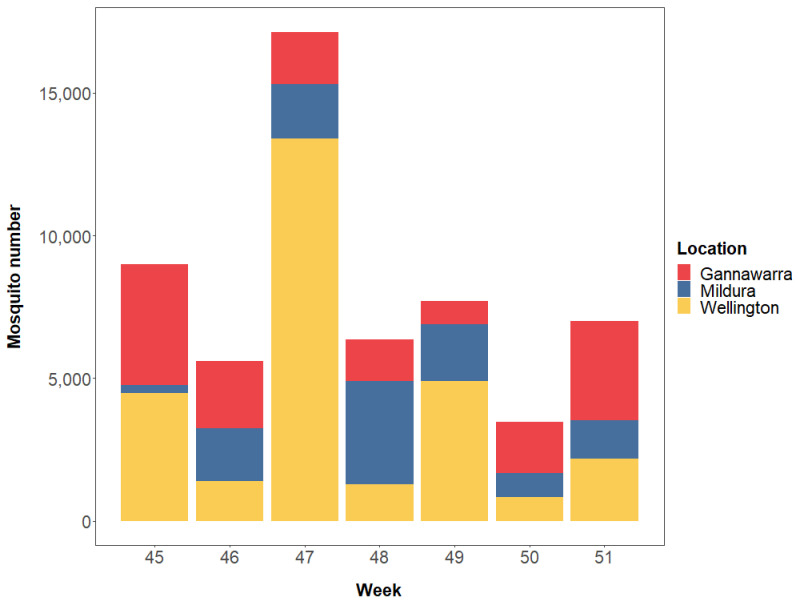
The estimated total number of mosquitoes used for sequencing per location per week from 7 November 2016 (Week 45) to 19 December 2016 (Week 51).

**Figure 3 viruses-14-02759-f003:**
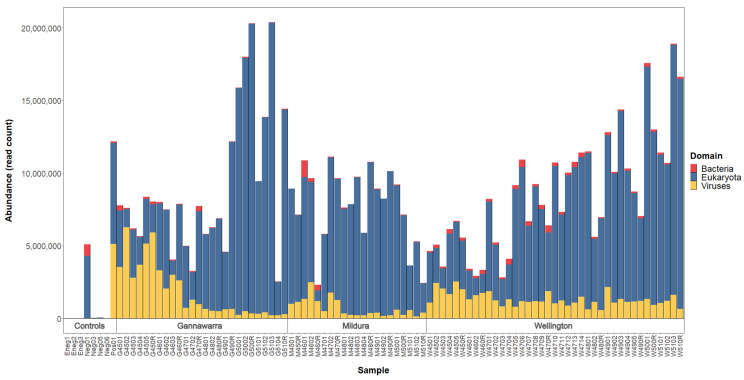
Domain-level taxonomic composition of the individual mosquito trap samples and control samples. Abundance is based on read counts for taxonomically classified contigs produced by *de novo* assembly. Archaea taxa were excluded from the figure due to their low abundance.

**Figure 4 viruses-14-02759-f004:**
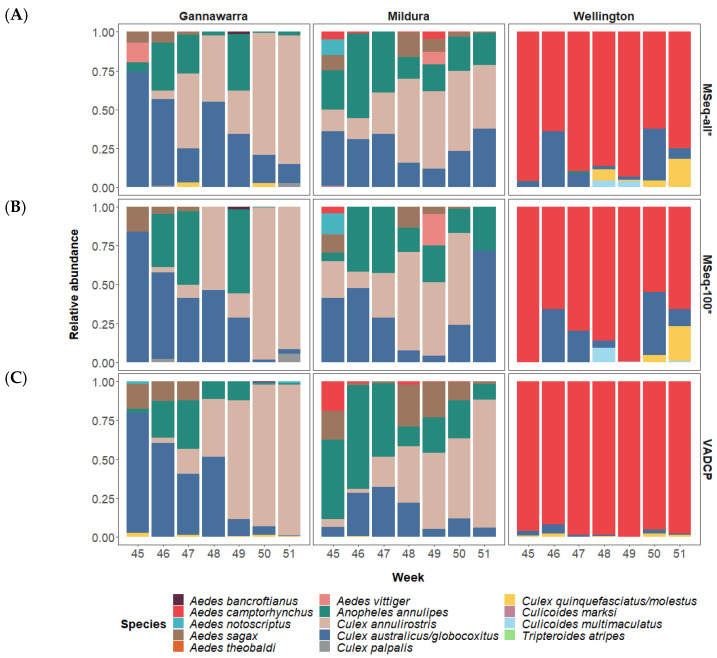
Mosquito and biting midge species identification in traps from Gannawarra, Mildura and Wellington over seven weeks (45–51) in 2016. Relative abundance estimations derived from a COI-based analysis of metatranscriptomic sequencing (MSeq) are shown in (**A**) when using all subsamples and (**B**) when using only the 100-mosquito subsamples. Members of the *Culex pipiens* species complex that cannot be differentiated by COI have been conglomerated. (**C**) shows species abundance based on morphological identification of mosquitoes subsampled from each trap by the Victorian Arbovirus Disease Control Program (VADCP) during the 2016/17 season. It should be noted that the VADCP does not survey biting midge species. The * indicates that (**A**,**B**) represent a separate mosquito subsample to (**C**).

**Figure 5 viruses-14-02759-f005:**
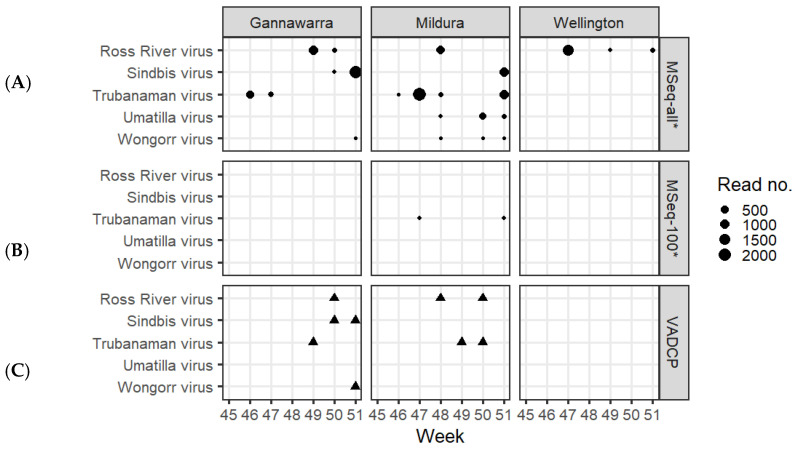
Arbovirus detections in traps from Gannawarra, Mildura and Wellington over seven weeks (45–51) in 2016. Detections based on metatranscriptomic sequencing (MSeq) are shown in (**A**) when using all subsamples and (**B**) when using only the 100-mosquito subsamples. The size of the dots is proportional to the number of reads contributing to each arbovirus detection. (**C**) shows the arboviruses detected via cell culture during the initial screening of mosquitoes subsampled from each trap by the Victorian Arbovirus Disease Control Program (VADCP) during the 2016/17 season. Each triangle represents one cell culture detection of each arbovirus. The * indicates that (**A**,**B**) represent a separate mosquito subsample to (**C**).

**Figure 6 viruses-14-02759-f006:**
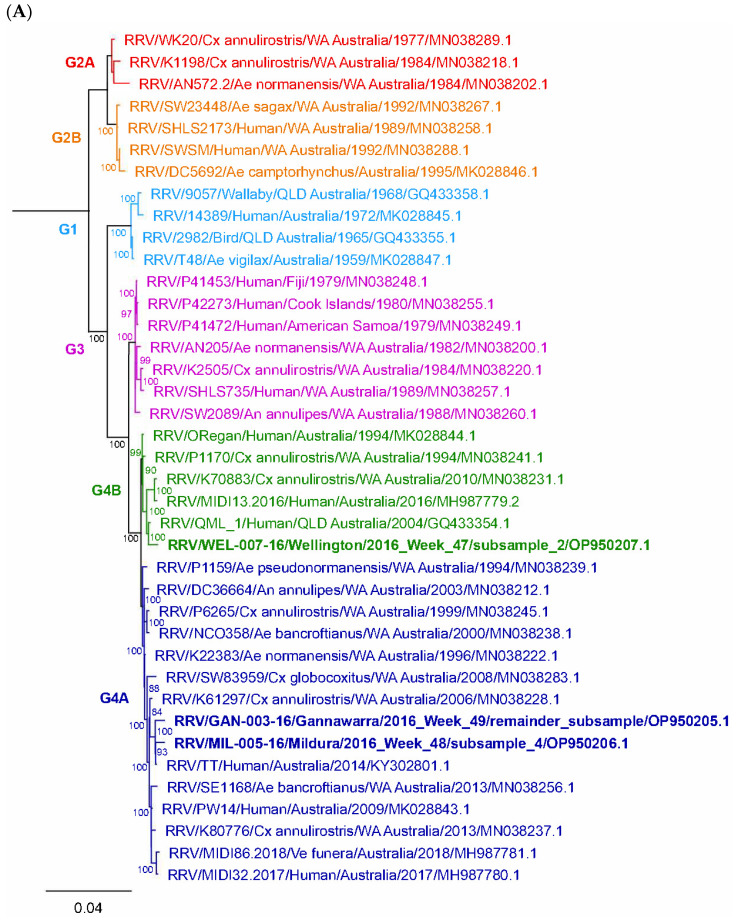
Phylogenetic trees depicting the genetic relationship of the (**A**) Ross River virus (RRV), (**B**) Sindbis virus (SINV), (**C**) Trubanaman virus (TRUV) and (**D**) Umatilla virus (UMAV) detections. The trees are based on whole genome sequence alignments for RRV (11,362 nt) and SINV (11,460 nt), Segment M for TRUV (4152 nt), and Segment 7 for UMAV (1364 nt). A maximum likelihood (GTR model) analysis was used with 1000 bootstrap replicates (only values > 70% were shown). Coloured clades represent virus genotypes, as indicated by the clade label. The naming convention for other viruses is virus/strain/host/location/year/GenBank accession, with boldface indicating sequences generated in this study.

**Figure 7 viruses-14-02759-f007:**
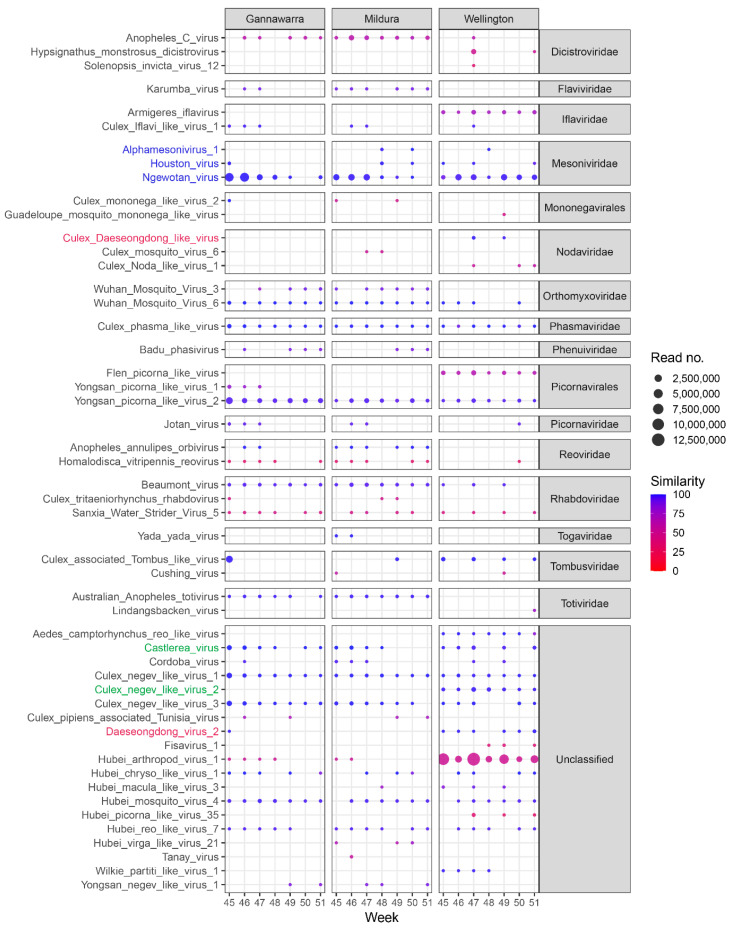
Results of the virome analysis showing the BLASTx match for contigs assembled from traps in Gannawarra, Mildura and Wellington over seven weeks (45–51) in 2016. The dot colour indicates the similarity of the contigs to the virus GenBank sequence, as measured by the percentage of identical amino acids. The dot size indicates virus abundance, which is based on the number of reads mapping to the matching contigs. Coloured names indicate highly similar or identical viruses (Appendix A) that likely represent a single detection. The arboviruses detected using the targeted database (Figure 5) are not included here.

**Figure 8 viruses-14-02759-f008:**
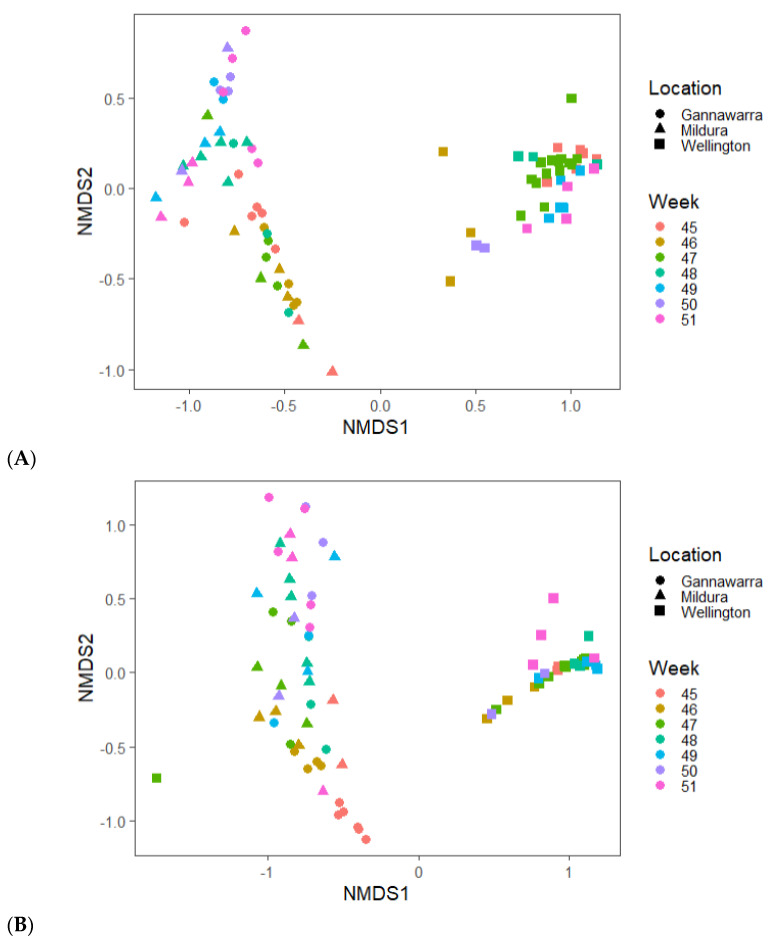
Non-metric multidimensional scaling (NMDS) based on Bray–Curtis dissimilarity of the (**A**) viruses and (**B**) mosquitoes in the trap subsamples from three locations over seven weeks showing a separation between coastal (Wellington) and inland (Gannawarra and Mildura) samples.

**Table 1 viruses-14-02759-t001:** Recommendations for the incorporation of metatranscriptomics in routine arbovirus surveillance programs.

Category	Recommendation	Outcomes
Sampling	Sequence all the mosquitoes collected from surveillance traps.	Increased likelihood of detecting arboviral activity.
	Investigate other sample types for metatranscriptomic sequencing (e.g., gravid mosquitoes, FTA cards)	Less sequencing of uninfected mosquitoes, thereby improving arbovirus detection sensitivity.
Laboratory protocol	Develop an in-house ribosomal RNA depletion protocol.	Greater ability to customise depletion, thereby improving arbovirus detection sensitivity; decreased assay cost.
	Use unique dual indexing to multiplex samples.	Reduced index cross-talk, thereby improving arbovirus detection sensitivity.
	Automate library preparation where possible.	Increased multiplexing capacity; decreased assay cost and turnaround time.
	Use ultra-high-throughput sequencing (e.g., Illumina NovaSeq).	Increased multiplexing capacity; decreased assay cost; faster turnaround than HiSeq.
Bioinformatics	Establish a user-friendly bioinformatics pipeline.	Less reliance on specialised bioinformatics expertise.
	Develop a tool to assess the risk of novel or unclassified viruses.	Faster, less complex data analysis with relevant reporting for public health.
	Formulate an organised and cost-effective storage plan for high volumes of sequencing data.	Ability to repurpose or re-analyse past metatranscriptomic surveillance data with updated databases and bioinformatics.
Reference databases	Establish a DNA barcode database of local mosquito species.	Comprehensive identification of mosquito species in surveillance traps.
	Acquire whole genome sequences of arbovirus isolates for inclusion in databases used for screening data.	Improved arbovirus detection sensitivity; high-resolution phylogenetics to determine arbovirus origins and dispersal.
	Curate a contamination database by sequencing samples from laboratory surfaces and reagents.	Improved ability to distinguish real signal from background or contamination.
Quality control	Include negative and positive controls.	Detection of contamination; ability to assess assay validity.
	Standardise laboratory and bioinformatics protocols.	Consistent, reproducible surveillance results of known sensitivity and specificity.
	Regularly validate assay sensitivity and specificity in response to protocol modifications.	Enables protocol updates while ensuring adequate assay sensitivity and appropriate detection thresholds.
	Confirm important arbovirus detections with RT-qPCR.	Confidence in reliability of arbovirus detections for public health reporting.

## Data Availability

The unprocessed Illumina HiSeq FASTQ read files for all 86 trap subsamples have been deposited into the NCBI SRA database (BioProject ID PRJNA642916). The custom nucleotide database used for the COI-based identification of mosquito and biting midge species is available on Figshare (https://doi.org/10.6084/m9.figshare.10246826.v3, accessed on 20 July 2022), as is the Australian arbovirus database (https://doi.org/10.6084/m9.figshare.12055830, accessed on 20 July 2022) – both accessed on 31 March 2020. The RRV, SINV, TRUV and UMAV sequences used for phylogenetic analysis are available online (GenBank acc. OP950205–OP950214), as are the longest contig sequences assembled for each virus detected as part of the broader virome analysis (https://doi.org/10.6084/m9.figshare.17071970, accessed on 20 July 2022).

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
