# Peer review of "Enhanced Arbovirus Surveillance with High-Throughput Metatranscriptomic Processing of Field-Collected Mosquitoes"

_viruses, 2022, doi:10.3390/v14122759_

Round 1
Reviewer 1 Report
This is a nice article that deserves to be published. I have listed below my comments which are meant to help clarify some points in the manuscript before its publication. It would have been nice to have line numbers, I did my best to clearly show which section I was referring to.
Material and methods : « enzootic and epizootic zones » : clarify if the difference comes from the regions or if this was done in all 3 regions
Virome analysis : it is not clear how you obtained the list from Figure 8 (and Figure S3). Was it the results from BLASTn of all contigs, or BLASTx or >1000nt long contigs ? what is the minimum size contig you considered ? Was the cut off an evalue of 10^-5 in all cases ?
I find it hard to understand how you got that list. For example in the case of the dicistroviridae, since your contigs have low similarity to the BLAST reference, there must have been other dicistrovirus species appearing on the BLAST results with low evalues, how did you pick/select which one to take ? Is there another criteria of % identity/similarity that was applied ?
I have similarly queries for the flavivirus Karumba virus, only ever detected in Anopheles meraukensis mosquitoes, which is not listed in your mosquito collection, and part of a group of viruses thought to be extremely specific to one mosquito species each. Are you reporting the detection of Karumba virus in Anopheles annulipes (the only Anopheles species in your list) or even another mosquito genus, or are you reporting the detection of a related virus, potentially in Anopheles annulipes ? And in this case, is the sequence more related to Karumba virus or to Haslams Creek virus, which is related to Karumba virus but was detected in Anopheles annulipes from Sydney ?
These are specific questions but I am asking them so you see why I want to understand how that list was curated and how your method can be used in the future for insect virome surveillance and potentially virus discovery.
Table 1 : it would be nice to have a visual representation rather than a table. For example sets of 3 pie charts (1 set per week, 1 pie chart per location), with pie sizes depending on the total mosquito count, so we can compare mosquito numbers in each region on the same week. You could still include the numbers in there too.
Upon reading further I see some of this info is in Figure 2. If you have already tried to combine these representations and it is too busy that’s fine, but if you haven’t I think it’s worth a try. Otherwise I am wondering if Table 1 would be a better fit in supplementary, the main information is on Figure 2 and in the percentages you give in the text, 9.6% on cell culture and 90.4% in metagenomics.
Figure 3 : I’m not sure I understand the black lines in each bar, are they log graduations ? They make the graph hard to read : some colours are hard to see and the figure as a whole looks busy. I also don’t see any archea bars, are they too small to see ? The archea colour is very close to that of the eukaryotes, consider changing it for clarity particularly for colourblind readers.
Metatranscriptomic mosquito and biting midge species identification : « The 12 mosquito species were detected over the seven-week period, with the two inland locations, Gannawarra and Mildura, sharing similar mosquito species in comparison to the coastal location, Wellington, which primarily had Aedes camptorhynchus » This is not clear. It is clear with the figure but on its own it sounds like all three locations have the same mosquitoes.
Figure 6 : These data would fit better near the metagenomic data they are being compared to. Fig6A should go as a panel in Fig4 and Fig6b should go in Fig5.
Was there only one cell culture detection of each virus per triangle symbol on Figure 6B ?
Phylogenetic analysis of arboviruses : it would be good to include a comment – possibly in the discussion if you don’t have the data – to say whether a better consensus/whole genome sequence can be generated from samples where the viruses have been detected after amplification in cell culture.
Virome ecology : Include Genbank accession numbers at the end of the paragraph on page 8.
« the pattern of abundance for Ngewotan virus resembles that of Culex australicus/globocoxitus » how did you notice this ? By eye ? is there a way to get this information in a more controlled way ?
« When the virus and mosquito species were compared in the three locations, the species found in inland Gannawarra and Mildura were similar compared to the species found in coastal Wellington » Same comment as previously it is not clear what you are comparing/contrasting, rephrase for clarity
Discussion : add accession numbers instead of « (acc xxx-xxx) »
Data availability statement : add accession numbers instead of « (acc xxx-xxx) »
Author Response
This is a nice article that deserves to be published. I have listed below my comments which are meant to help clarify some points in the manuscript before its publication. It would have been nice to have line numbers, I did my best to clearly show which section I was referring to.
We thank the reviewer for their thoughtful and helpful comments. We would like to note that although the submitted manuscript had line numbers, they must have been removed during the submission process, and we apologise for this inconvenience.
Material and methods: « enzootic and epizootic zones »: clarify if the difference comes from the regions or if this was done in all 3 regions
Clarification of where the different traps were located has been added.
Virome analysis: it is not clear how you obtained the list from Figure 8 (and Figure S3). Was it the results from BLASTn of all contigs, or BLASTx or >1000nt long contigs? what is the minimum size contig you considered? Was the cut off an evalue of 10^-5 in all cases?
For the sake of brevity, the viruses listed in Figure 8 (now Figure 7) are based on >1,000nt contigs with a minimum of 100 reads that matched viral RdRp sequences in RefSeq with an e-value <10-5 and an alignment length >500bp. Figure S3 is a phylogenetic tree comparing the GenBank RdRp sequences that matched the contigs to see if there were any highly similar or identical RdRp references that may have caused redundancies in virus detection during the analysis. These details have been added to the virome analysis methods, which have also been restructured to make it clear how the analysis was performed.
I find it hard to understand how you got that list. For example in the case of the Dicistroviridae, since your contigs have low similarity to the BLAST reference, there must have been other dicistrovirus species appearing on the BLAST results with low evalues, how did you pick/select which one to take? Is there another criteria of % identity/similarity that was applied?
All Dicistroviridae viruses that had RefSeq RdRp sequences matching >1,000nt contigs with a minimum of 100 reads, an e-value <10-5 and an alignment length >500bp were included in the figure, and these criteria are now included in the virome analysis methods. There were other contigs with matches to Dicistroviridae viruses but they did not meet the above criteria, including taxa that were not part of GenBank’s RefSeq database. Some of these contigs likely represent real detections of novel viruses, along with many other contigs outside of these narrow criteria. However, a thorough characterisation of the mosquito virome was outside the scope of this study, which is why we have uploaded both the longest contig assembled for each virus included in the figure and the Illumina reads for each trap subsample, so that others may further investigate these results.
I have similar queries for the flavivirus Karumba virus, only ever detected in Anopheles meraukensis mosquitoes, which is not listed in your mosquito collection, and part of a group of viruses thought to be extremely specific to one mosquito species each. Are you reporting the detection of Karumba virus in Anopheles annulipes (the only Anopheles species in your list) or even another mosquito genus, or are you reporting the detection of a related virus, potentially in Anopheles annulipes? And in this case, is the sequence more related to Karumba virus or to Haslams Creek virus, which is related to Karumba virus but was detected in Anopheles annulipes from Sydney? These are specific questions but I am asking them so you see why I want to understand how that list was curated and how your method can be used in the future for insect virome surveillance and potentially virus discovery.
The contigs matching Karumba virus had ~90% amino acid and ~79% nucleotide identity, indicating the presence of a closely related novel virus. Haslams Creek virus is not part of the RefSeq database and so would not have come up as the closest RdRp match. However, even if the BLASTx RdRp search was extended to all of nr, Karumba virus would have still been the closest match, as these contigs only have ~76% amino acid and ~86% nucleotide identity to Haslams Creek virus. Additional details have been added to the virome analysis methods to better describe how the list of virus matches was attained. We have also uploaded the longest contig for each virus match so that others may further investigate the mosquito virome results that were outside the scope of this study, which is discussed in the second paragraph on Page 17.
Table 1: it would be nice to have a visual representation rather than a table. For example sets of 3 pie charts (1 set per week, 1 pie chart per location), with pie sizes depending on the total mosquito count, so we can compare mosquito numbers in each region on the same week. You could still include the numbers in there too. Upon reading further I see some of this info is in Figure 2. If you have already tried to combine these representations and it is too busy that’s fine, but if you haven’t I think it’s worth a try. Otherwise I am wondering if Table 1 would be a better fit in supplementary, the main information is on Figure 2 and in the percentages you give in the text, 9.6% on cell culture and 90.4% in metagenomics.
As the most essential mosquito count information is available in the text and Figure 2, Table 1 has been moved to Supplementary Materials.
Figure 3: I’m not sure I understand the black lines in each bar, are they log graduations? They make the graph hard to read: some colours are hard to see and the figure as a whole looks busy. I also don’t see any archaea bars, are they too small to see? The archaea colour is very close to that of the eukaryotes, consider changing it for clarity particularly for colourblind readers.
The black lines in each bar represented taxa OTUs in each domain – these have now been removed from Figure 3 to better illustrate the proportion of each domain in each sample. Additionally, the archaea domain has been removed from the figure as it was represented by only two taxa with very low abundance that were not visible in the bars. The Figure 3 caption has been updated to reflect this change.
Metatranscriptomic mosquito and biting midge species identification: « The 12 mosquito species were detected over the seven-week period, with the two inland locations, Gannawarra and Mildura, sharing similar mosquito species in comparison to the coastal location, Wellington, which primarily had Aedes camptorhynchus » This is not clear. It is clear with the figure but on its own it sounds like all three locations have the same mosquitoes.
The word ‘comparison’ has been changed to ‘contrast’ in order to better describe the figure.
Figure 6: These data would fit better near the metagenomic data they are being compared to. Fig6A should go as a panel in Fig4 and Fig6b should go in Fig5.
Figure 6A has been added to Figure 4 and Figure 6B has been added to Figure 5.
Was there only one cell culture detection of each virus per triangle symbol on Figure 6B?
Yes, there was only one cell culture detection for each arbovirus per triangle symbol on Figure 6B (now Figure 5C) and the figure caption has been updated to reflect this.
Phylogenetic analysis of arboviruses: it would be good to include a comment – possibly in the discussion if you don’t have the data – to say whether a better consensus/whole genome sequence can be generated from samples where the viruses have been detected after amplification in cell culture.
As all of the virus sequences generated as part of this study were derived from uncultured material, we cannot provide any data to show improved genome sequencing after amplification via cell culture. However, a comment has been added to the discussion on Page 14 to suggest cell culture could be used in future studies to improve whole genome sequencing.
Virome ecology: Include GenBank accession numbers at the end of the paragraph on page 8.
We are currently waiting for accession numbers to be assigned to data uploaded as part of this manuscript and expect that they will be ready in time for publication.
« the pattern of abundance for Ngewotan virus resembles that of Culex australicus/globocoxitus » how did you notice this? By eye? is there a way to get this information in a more controlled way?
Further investigation of the abundance patterns for Ngewotan virus and Culex australicus/globocoxitus in the trap subsamples indicated a statistically insignificant correlation and so this sentence has been removed from the results section.
« When the virus and mosquito species were compared in the three locations, the species found in inland Gannawarra and Mildura were similar compared to the species found in coastal Wellington » Same comment as previously it is not clear what you are comparing/contrasting, rephrase for clarity.
The second ‘compared’ has been changed to ‘in contrast’ to better describe the figure.
Discussion: add accession numbers instead of « (acc xxx-xxx) »
Data availability statement: add accession numbers instead of « (acc xxx-xxx) »
We are currently waiting for accession numbers to be assigned to data uploaded as part of this manuscript and expect that they will be ready in time for publication.
Reviewer 2 Report
This manuscript is of good scientific reference value. However, the identification of mosquito species is rough due to the lack of morphological identification data and only COI sequence data. 2. Lack of exact culture and isolation data of virus species, if only 1-2 types of evidence is good. Overall, this manuscript is based on environmental transcriptome analysis data, and the data are still novel. It is suggested to be published after minor modification.
Author Response
This manuscript is of good scientific reference value. However, the identification of mosquito species is rough due to the lack of morphological identification data and only COI sequence data.
We agree with the reviewer that morphological identification of the specimens used for sequencing would have been ideal to validate the COI-based metatranscriptomic mosquito species identification results. However, the resources needed to morphologically identify over 56,000 mosquitoes were not within the bounds of this study. Furthermore, the mosquito species abundances acquired by the VADCP in 2016 using morphological methods corresponded to the COI-based abundance estimates, despite being based on separate subsamples of the same traps. We have combined the COI-based and morphology-based mosquito species abundance estimates into one figure (Figure 4) to better demonstrate how the metatranscriptomic species identification was supported by the VADCP data.
- Lack of exact culture and isolation data of virus species, if only 1-2 types of evidence is good.
All detections of arboviruses of public health significance were further investigated using RT-qPCR, with 31 out of 33 detections confirmed. As for the 51 viruses detected via de novo assembly, the resources required to culture and isolate these viruses were outside the bounds of this study. All of these additional viruses are considered insect-specific or have no known vertebrate host, so further investigation and characterisation of these viruses are outside the scope of this study. The longest contig assembled for each of the 51 viruses and the Illumina reads for each trap subsample have been made publicly available so that others may further investigate these results.
Overall, this manuscript is based on environmental transcriptome analysis data, and the data are still novel. It is suggested to be published after minor modification.
We thank the reviewer for their valuable comments and believe the modifications made to the revised manuscript address all primary concerns.